# Three-Phase Partitioning for the Extraction and Purification of Polysaccharides from the Immunomodulatory Medicinal Mushroom *Inonotus obliquus*

**DOI:** 10.3390/molecules24030403

**Published:** 2019-01-22

**Authors:** Zhendong Liu, Dongsheng Yu, Liang Li, Xiaoxiao Liu, Henan Zhang, Wenbo Sun, Chi-Chung Lin, Jiafu Chen, Zhi Chen, Wenhan Wang, Wei Jia

**Affiliations:** 1Food Science College, Tibet Agriculture & Animal Husbandry University, Linzhi 860000, China; liu304418091@126.com (Z.L.); jwllok@sina.com (L.L.); 18889040987@sina.com (J.C.); 2Institute of Edible Fungi, Shanghai Academy of Agricultural Sciences, Key Laboratory of Edible Fungi Resources and Utilization (South), Ministry of Agriculture, P.R.China; National Engineering Research Center of Edible Fungi, National R&D Center for Edible Fungi Processing, Shanghai 201403, China; m18706106105@163.com (D.Y.); 18369610051@163.com (X.L.); henanhaoyun@126.com (H.Z.); ccl92122@yahoo.com (C.-C.L.); 3College of Food Science and Technology, Shanghai Ocean University, Shanghai 201306, China; 4Shandong Provincial Key Laboratory of Animal Disease Control & Breeding, Animal Science and Veterinary Medicine Institute of Shandong Academy of Agricultural Sciences, Jinan 250100, China; sunwenboo@gmail.com (W.S.); charleschenzhi@163.com (Z.C.)

**Keywords:** *Inonotus obliquus*, three-phase partitioning, free-radical scavenging abilities, antioxidant activity, immunological activity

## Abstract

Polysaccharides from the immunomodulatory medicinal mushroom *Inonotus obliquus* (IOPS) were extracted and purified using three-phase partitioning (TPP), which is an efficient, fast, safe, and green purification technique. An optimal extraction procedure that gave a good 2.2% isolated yield was identified, using the following protocol: a solid-liquid ratio of 1 g to 12 mL; mass fraction of (NH_4_)_2_SO_4_ 20% (*w*/*v*); 11 mL *t*-butanol; pH 8.0; temperature 30 °C; and extraction time 30 min. The purified IOPS was shown to be a proteoglycan of 40 kDa molecular weight comprising of d-galactose, d-glucose, d-xylose, and d-mannose in a molar ratio of 2.0:3.5:1.0:1.5. The purified IOPS displayed strong free-radical scavenging abilities, antioxidant activities, and immunological activity in vitro. IOPS’ Trolox antioxidant equivalent capacity and ferric-reducing ability of plasma were 251.2 μmol Trolox/g sample and 1040.5 μmol Fe^2+^/g sample, respectively, with the activity of its immunomodulatory behavior shown to be gradient dependent.

## 1. Introduction

*Inonotus obliquus* (Ach. ex Pers.) Pilát, also known as chaga, is a medicinal basidiomycetes fungus included in the Hymenochaetaceae family, Hymenochaetales order, and Agaricomycetes class, respectively [1]. *I. obliquus* is extremely cold-resistant, with its mycelium growing in woods that tolerates temperatures as low as −40 °C. It is normally present at latitudes of 45–50°. It occurs widely in Russia, North Europe, Poland, the Hokkaido region of Japan, Heilongjiang province, and the Changbai mountain area in the Jilin province of China. Because of the special conditions required for its growth, indigenous people have variously described these fungi as “black diamonds in the forest”, “fantastic mushrooms”, and other namesakes [2]. *I. obliquus* has attracted increasing attention worldwide due to its high nutritional and medicinal values, as well as its anti-tumor [3], anti-oxidant [4], immunomodulatory activities [5], anti-asthma [6], and other medicinal properties [7]. Several studies have shown that polysaccharides are one of the major bioactive constituents that are present in fresh *I. obliquus* [8]. Many researchers developed protocols for the extraction of edible fungal polysaccharides. The main extraction methods employed include immersion extraction, ultrasonic extraction, microwave-assisted extraction, and high-voltage pulsed electric field-assisted extraction. Zhang et al. used ultrasonic assisted extraction to develop optimal conditions for the extraction of polysaccharides from *I. obliquus*, with the best conditions involving use of water as a solvent for 15 minutes at 95 °C with ultrasound frequency (20 KHz, 60 min), which gave extraction values of 1.82% [9]. However, these extraction methods are complex, requiring a large amount of solvent, high temperatures, high acoustic pressures, and long treatment times. These multiple purification and separation steps are difficult to carry out on a large-scale.

Three-phase partitioning (TPP) is a new protein and edible oil separation and purification technology, which combines salting out, isoelectric point precipitation, and solvent precipitation processes. A protein sample is treated with an organic solvent (normally *t*-butanol) and then a salt solution, and the resultant mixture is stirred and centrifuged to afford a three phase mixture comprised of an upper organic phase, a lower aqueous phase, and a central protein phase [10]. TPP is often formed by adding water miscible aliphatic alcohol and salt to a slurry of protein, forming an alcohol-rich upper phase, solid middle phase of protein, and salt-rich lower phase [11,12,13]. The pigments, lipids, and hydrophobic materials are concentrated in the upper phase, the protein is concentrated in the middle phase, saccharides and other polar components are concentrated in the lower phase. This TPP technique selectively separates the desired protein into the mesophase, with any remaining free protein distributed between the other two phases. This approach has been widely used for the isolation of commercially valuable proteins and oil. Sharma used TPP to purify pectinases from *Aspergillus niger* and tomato, with the yields of 76% (*Aspergillus niger*) and 183% (tomato) and purifications of 10-fold (*Aspergillus niger*) and 9-fold (tomato) [14]. Kulkarni used TPP, enzyme, and supercritical carbon dioxide extraction (SCE) to extract linseed oil, the enzyme-pre-treated TPP using accellerase is recommended due to excellent protein recovery of 86.62%, and better oil quality [15].

Currently, TPP is also applied to extract and purify tapioca starch and tapioca starch derivatives [16], chitosan from shrimp shell [17], alginates from *Dunaliella salina* [18], levan, and hydrolyzed levan from several levansucrase microorganisms, among them *Zymomonas mobilis*, a mobile Gram-negative bacterium [19], aloe polysaccharides [20], and *Corbicula fluminea* polysaccharides [21]. More recently, Wang et al. reported that TPP is utilized to separate and purify polysaccharide–protein complexes (PSP) from *C. fluminea* [22]. The highest extraction yield of PSP was 9.0% under the following optimal conditions: 20% (*w*/*v*) ammonium sulfate concentration, 1.5: 1.0 (*v*/*v*) *t*-butanol to crude extract ratio, 30 min, and 35 °C. The purified PSP also exhibited strong radical scavenging capacities and antioxidant activities in vitro. TPP have been applied in the fields of plants, animals, and microorganisms. However, its use for the separation of polysaccharides from edible and medicinal fungi has not yet been reported.

This report describes the development of a TPP process for the efficient extraction and separation of polysaccharides from *I. obliquus* (IOPS), using a response surface method that gave a 2.1% extraction yield (EY) of polysaccharides with a defined molecular weight of around 40 kDa that was shown to display high antioxidant and immunological activities in vitro.

## 2. Results and Discussion

### 2.1. Optimization of TPP Conditions

The most important factors that affect the three-phase purification of a polysaccharide are its physical properties, its concentration, the organic solvent used for extraction, pH, the type and concentration of inorganic salt present, and the length and temperature of the extraction process.

#### 2.1.1. Effect of Organic Solvent and Solid-Liquid Ratio on the Extraction Process

The organic solvent not only affects the polarity and dielectric constant of the extraction process, but also has an important influence on the formation and stability of the three-phase system. Therefore, selecting an appropriate organic solvent plays an important role in the purification process of TPP [23]. As shown in Figure 1a, when *t*-butanol was used as the organic solvent for the three-phase system, an IOPS EY of up to 1.83% was obtained, whilst EY of IOPS using other organic solvents were lower. This may be due to the molecular weight and branched structure of *t*-butanol preventing it from penetrating into the folded tertiary structure of the protein resulting in an effective TPP protocol [24]. Other organic solvents tended to denature and inactivate the enzyme protein, making it more difficult to form the protein and aqueous phases of the TPP system. Therefore, *t*-butanol was selected as a solvent to carry out the three-phase separation and purification of IOPS.

Figure 1b shows the relationship between the solid-liquid ratio and the polysaccharide extraction rate for the crude polysaccharide extract of *I. obliquus*. The extraction rate of IOPS was optimal when the ratio (mL to g) of material to liquid was 1:12, with poorer extraction efficiency and greater reagent waste observed when the solid-liquid ratio was 1:6. 1:24, 1:48, respectively.

#### 2.1.2. Effects of Amount of *t*-Butanol, Mass Fraction of (NH_4_)_2_SO_4_, Temperature, pH and Time on the Extraction Process

It is known that *t*-Butanol favors formation of organic copolymers, where it can combine with ammonium sulfate to separate organic impurities, such as enzyme proteins and pigments from a crude extract into the *t*-butanol phase. Therefore, the amount of *t*-butanol employed in the extraction process can significantly affect the efficiency of three-phase methods [25]. As shown in Figure 2a, the EY of IOPS increases with increasing amounts of *t*-butanol, with the highest extraction rate for IOPS observed using 15 mL of *t*-butanol.

The mass fraction of (NH_4_)_2_SO_4_ employed in the extraction process is closely related to formation of a protein precipitate, with increasing mass fractions of (NH_4_)_2_SO_4_ normally resulting in greater amounts of protein precipitating out of aqueous phase due to the salting out effect [26]. As shown in Figure 2b, when the mass fraction of ammonium sulfate was changed from 10% to 60% (*w*/*v*), the EY of IOPS initially increased and then decreased, with a maximum EY of 2.2% observed at 20% (*w*/*v*). This is due to IOPS being more effectively distributed when ammonium sulfate salt is present at a low concentration, with higher ammonium sulfate content resulting in hydrogen bonds between IOPS and water molecules being disrupted, which leads to a decreased extraction rate.

Figure 2c shows that the amount of IOPS isolated increased between 15 and 30 °C, with temperatures of >30 °C leading to a gradual decrease in EY of IOPS. Formation of a three-phase system resulted in more *t*-butanol crystallization occurring, which affected mass transfer efficiency that led to slower three-phase formation. Increasing temperature results in *I. obliquus* becoming more hydrophilic, because the hydroxyl groups of its more extended structure are more exposed and capable of forming greater numbers of hydrogen bonds to solvent molecules [27]. The EY of IOPS reached its highest level at 30 °C, which was chosen as the optimum temperature for isolating IOPS.

The ability of pH to change the electrostatic charge of a free protein in the aqueous phase is well known [28], which can affect EY of polysaccharide in the aqueous phase. As shown in Figure 2d, when the pH was changed from 4.0 to 5.0, the extraction rate of IOPS increased gradually. However, the extraction rate decreased between 5.0 to 6.0, and then increased again from 6.0 to 8.0. A maximum extraction value of around 1.82% was achieved at pH = 8.0, with mildly acidic conditions resulting in some of the acidic polysaccharides and glycoproteins remaining in the aqueous phase.

Extraction time is a crucial factor in the formation of a three-phase system, with extended time contributing towards increased production costs. Figure 2e shows that an extraction time of 30 min gave a maximal IOPS EY, with further increases in extraction time having little effect. Therefore, 30 min was chosen as an appropriate time for the three-phase extraction of IOPS.

### 2.2. Optimization of TPP Extraction Parameters by RSM

The most important physical factors that affect the EY of IOPS are the mass fraction of ammonium sulfate (A, %), the amount of *t*-butanol (B, mL), and the temperature (C, °C), with suitable levels for these parameters determined using statistical central composite design (CCD). The experimental design matrix that was used is shown in Table 1, with seventeen experiments performed using different combinations of variables for CCD. The results of these experiments were used as the basis of the following second order polynomial equation to calculate the EY of IOPS as a function of the mass fraction of ammonium sulfate (A, %), amount of *t*-butanol (B, mL), and temperature (C, °C):Y = 2.201792455 + 0.0075 A + 0.00877303 B + 0.015668969 C − 0.009 A B + 0.01 A C − 0.008662061 B C − 0.126669257 A^2^ − 0.244123198 B^2^ − 0.135123198 C^2^(1)

The data from Table 1 reveals a model F value of 32.91 with *p* < 0.0001, indicating that the regression model was very significant, with the model determination coefficient R^2^ = 0.9769 indicating that only 2.31% of the total variation could not be explained by the model. These values indicate that the model is a good fit, with the small *P*-value suggesting a high significance level for the corresponding coefficient [29]. The loss of the proposed item *p* > 0.05 reveals that the insignificant item and the impact were not significant, once again indicating that the selected model was appropriate. The A^2^, B^2^, and C^2^ values for the extraction of IOPS were very significant. However, all other factors were not significant.

Figure 3 reveals the isoresponse contour and surface plots that were used to optimize the extraction conditions to maximize EY of IOPS. The EY of 2.10% was predicted for a 1 g: 12 mL ratio of solid to liquid, a constant pH value of 8.0, an extraction time of 30 min, an ammonium sulfate mass fraction of 20.16%, a volume ratio of *t*-butanol of 1:1.15, and a temperature of 30.3 °C. The reliability of this response surface model was verified by carrying out three replicate experiments using the same conditions, which gave an average EY for IOPS of 2.13 ± 0.12% that was close to the predicted value. Therefore, the response surface model established in this study is suitable for optimizing the three-phase extraction of IOPS.

### 2.3. Physicochemical Properties and Bioactivities of IOPS In Vitro

#### 2.3.1. Characterization of the Physicochemical Properties of IOPS

Extracted IOPS from the lower aqueous phase was subjected to dialysis, and lyophilized to afford purified IOPS, with its overall purity increasing from 10.31% in the crude extract to 57.17% after TPP (including dialysis). Total carbohydrate and protein contents were estimated to be 57.17% and 32.53%, respectively.

HPLC analysis of the purified IOPS produced a single symmetrical peak, indicating that the sample was a homogeneous polysaccharide (Figure 4a). The molecular weight and chain conformation of the purified IOPS were determined using (high performance size exclusion chromatography multiangle laser light scattering) HPSEC-MALLS. The radius of gyration value (Rg), weight average molecular weight (Mw), and polydispersity index (Mw/Mn) of IOPS in PBS buffer were found to be 33.0 (±28.80%) nm, 4.05 × 104 (±9.08%) g/mol, and 1.25 (±14.44%), respectively (Figure 4b). Figure 5b shows that purified IOPS has a typical characteristic absorption peak of polysaccharide, a strong -OH stretching vibration peak at 3223 cm^−1^, and a strong C-H stretching vibration peak of -CH_3_, -CH_2_, -CH at 2963 cm^−1^, a C = O asymmetric stretching vibration peak at 1643 cm^−1^, CH variable angle vibration peak at 1419 cm^−1^. Three absorption peaks at 1088, 775, and 615 cm^−1^ indicated a pyranose characteristic of the ring [30]. Monosaccharide compositional analysis of the purified IOPS indicated the presence of d-galactose, d-glucose, d-xylose and d-mannose in a molar ratio of 2.0:3.5:1.0:1.5.

#### 2.3.2. Antioxidant and Immunological Activities of IOPS In Vitro

In this study, the DPPH radical scavenging activity, TEAC, and FRAP were used to determine the in vitro antioxidant activities of IOPS. Figure 5a shows the DPPH scavenging activities of purified IOPS after TPP compared with Vc, which show that purified IOPS exhibits a dose-dependent free radical scavenging capacity in a manner over a concentration range of 0-0.5 mg/mL. At 0.5 mg/mL, the DPPH scavenging activity of purified IOPS was 78%, which was lower than for Vc (0.02 mg/mL, 85.2%). As shown in Figure 5b, IOPS showing strong antioxidant action with TEAC values of 1040.5 μmol Fe^2+^/g sample and 251.2 μmol Trolox/g, respectively.

Production of NO after macrophages activation is regarded as an important signal transduction factor for the immune system, with reactive nitrogen intermediates (RNI) playing an important role on the capacity of macrophages to kill tumor cells [31]. The level of RNI produced in the presence of IOPS was estimated by determining the concentration of NO formed, which is known to be a relatively stable metabolite of RNI. As shown in Figure 5c, concentrations of IOPS between 50 and 500 μg/mL resulted in dose-dependent release of NO by RAW264.7 macrophages, affording values in excess of 500 μg/mL after 24 h.

## 3. Materials and Methods

### 3.1. Materials and Chemicals

*I. obliquus* fruiting bodies were collected from Tibet’s Linzi Sera Mountain Region and their identity confirmed by Professor Cui Baokai of Beijing Forestry University and Professor Xu Asheng of the Institute of Plateau Ecology, Tibet Agricultural and Animal Husbandry Institute. Raw264.7 cell lines were purchased from tCell Resource Center of Shanghai Institute of Life Sciences, Chinese Academy of Sciences (Shanghai, China). DMEM medium and fetal bovine serum (FBS) were purchased from Thermo Fisher Scientific Inc. (Waltham, MA, USA). The 1,1-diphenyl-2-picrylhydrazyl (DPPH), 6-hydroxy-2,5,7,8-tetramethylchroman-2-carboxylic acid (Trolox), 2,2′-azinobis-(3-ethylbenzothiazoline-6-sulphonic acid) (ABTS), and 2,4,6-tris (2-pyridyl)-s-triazine (TPTZ) were purchased from Sigma-Aldrich (St Louis, MO, USA). All other reagents were of analytical reagent grade and obtained from Sinopharm Chemical Reagent Co., Ltd. (Shanghai, China).

### 3.2. Sample Pretreatment

Fresh air-dried *I. obliquus* fruiting bodies were ground to powder and passed through a 10-mesh sieve. Powder of *I. obliquus* (50.0 g) was suspended in 500 mL of 95% ethanol in distilled water (*v*/*v*) and extracted for 1h at the frequency of 40 KHz using a SB25-12D ultrasonic generator (Ningbo Scientz Biotechnology Co., Ltd., Ningbo, China.) at room temperature to remove lipids. This operation was repeated three times. After filtration, the residue was air-dried at room temperature, suspended in 20 volumes of distilled water, and extracted twice for 2 h at 100 °C. The liquid extracts were combined, centrifuged (26,000× *g*, 20 min, 20 °C), and the supernatant was concentrated to 300 mL under vacuum that was labelled as the crude extract.

### 3.3. TPP

(NH_4_)_2_SO_4_ (10–60%, *w*/*v*) was added to the crude extract (10 mL) that was then vortexed gently, followed by addition of *t*-butanol (5–30 mL). The extraction tube was placed in a shaking incubator at 100 g/min for 15-90 min, and the mixture centrifuged (2770× *g*, 10 min, 20 °C) to afford three clear phases. The upper organic phase (*t*-butanol) was collected and recycled by evaporation under reduced pressure. Free proteins were found to be extracted almost exclusively into the middle phase [32]. The lower aqueous phase was mainly composed of (NH4)_2_SO_4_ and IOPS, which were collected dialyzed (cut-off at 8.0-10 kDa) for 4 days, concentrated to ~10 mL under vacuum at 45 °C, and then followed by lyophilization to give purified IOPS. The *n*-Butanol, isobutanol, *t*-butanol, *n*-amyl alcohol, and acetonitrile were all screened as solvents to determine the effect of using different organic solvents on the extraction yield of IOPS. Effects of various process parameters on the extraction yield of IPOS were evaluated, including the mass fraction of (NH_4_)_2_SO_4_ (%, *w*/*v*), the amount of *t*-butanol (mL), extraction times (min), temperature (°C), and pH. After the formation of three phases, each phase was carefully separated, the volume of the lower phase was noted, and the content of IOPS in the lower phase was determined.

The EY (%) of IOPS was represented by: EY = (CL VL)/M 100(2)
where M is the total mass of the original *I. obliquus* powder and CL and VL are the concentration and volume of IOPS in the lower phase at equilibrium, respectively.

### 3.4. Optimization of TPP Conditions by Response Surface Methodology (RSM)

Single factor experiments employing RSM with a 3^3^ (three-factor-three-level) factorial Box-Behnken design (BBD) were used to optimize the three-phase conditions used for extraction. Three factors, namely mass fraction of (NH_4_)_2_SO_4_ (X1, %), amount of *t*-butanol (X2, mL), and temperature (X3, °C) were chosen as independent (input) variables. The EY of IOPS (%) was chosen as the response value. According to the design principle of BBD response surface, the response surface analysis experiment was designed.

### 3.5. Preliminary Study on the Polysaccharide Composition of IOPS

IOPS was dissolved in an aqueous solution (2 mg/mL), and its homogeneity and molecular weight determined by high performance size exclusion chromatography (HPSEC). The analytical system consisted of a Waters 2695 HPLC system equipped with multiple detectors: a refractive index detector (RI), a UV detector to determine concentration, a multiple angle laser light scattering detector (MALLS, DAWNHELEOS, Wyatt Technology, Goleta, CA USA) for direct molecular weight determination, and a differential pressure viscometer (DP) for viscosity determination. Analysis was carried out using TSK PWXL 6000 and 4000 gel filtration columns that were eluted using PB buffer (0.15 M NaNO_3_ and 0.05 M NaH_2_PO_4_, pH = 7) at a flow rate of 0.5 mL/min. The column and RI detector temperature were maintained at 35 °C [33].

IOPS (2 mg) was hydrolyzed with 3 mL of 2 mol/L trifluoroacetic acid (TFA) at 110 °C for 4 h, and the solution then evaporated to dryness under reduced pressure. Monosaccharide composition was determined by high-performance anion-exchange chromatography (HPAEC) using a Dionex ICS500 equipped with a CarboPacTM PA20 column (3 × 150 mm). The column was eluted with 2 mmol/L NaOH (0.45 mL/min) and the monosaccharides detected using a pulsed amperometric detector (Dionex). d-Arabinose (d-Ara), d-glucose (d-Glc), d-glucosamine (d-GlcN), d-galactose (d-Gal), d-galacturonic acid (d-GalA), d-mannose (d-Man), l-rhamnose (l-Rha), d-xylose (d-Xyl), d-glucuronic acid (GlcA), and galacturonic acid (GalA) were used as monosaccharide standards [34].

Fourier transforms infrared (FT-IR) spectroscopic analysis of samples compressed in KBr pellets [35] were performed using a Nexus 670 FT-IR spectrometer (Thermo Nicolet Co., USA) operating over a wave number range of 500 to 4000 cm^−1^.

### 3.6. In Vitro Antioxidant Activity

#### 3.6.1. The DPPH Assay

The DPPH assay was carried out using modifications of the method described by Brand-Williams [36]. Stock solutions were prepared by dissolving 4 mg of DPPH in 100 mL ethanol, and then stored in the dark until required. Stock solution (or ethanol) (3 mL) was mixed with 1 mL of different concentrations of vitamin C (or sample), and then the absorbance was detected at 517 nm in a spectrophotometer:DPPH radical scavenging activity (%) = [1 − (A_1_ − A_2_)/A_0_] × 100%(3)
where A_1_ is the absorbance of the sample (or ascorbic acid) in the DPPH solution; A_2_ is the absorbance of a mixture of the sample in deionized water; A_0_ is the absorbance of a blank sample comprised of deionized water.

#### 3.6.2. Ferric Reducing Ability of Plasma (FRAP) Assay

The ability of IOPS samples to reduce ferric ions were measured using a modified version of a previously described method [37]. A sample (100 μL) was added to 900 uL of FRAP reagent (10 mM TPTZ solution, 20 mM FeCl_3_.6H_2_O solution, 300 mM acetate buffer at pH 3.6 (volume ratio of 1:1:10)), and the reaction mixture incubated at 37 °C for 2 h. After the end of the incubation period, the absorbance value of the sample was measured at 593 nm. The antioxidant capacities of the samples were determined based on reduction of ferric ions in these samples that were calculated from the linear calibration curve and expressed as mM FeSO_4_ equivalents per gram of sample.

#### 3.6.3. Trolox Equivalent Antioxidant Capacity (TEAC) Assay

The ABTS free radical scavenging activity of the samples were determined via modification of previously described methods [38]. A mixture of ABTS (7.4 mM) and potassium persulfate (4.95 mM) was shaken overnight at room temperature in the dark to form the free radical cation ABTS^+^·. A working solution was diluted with phosphate buffer solution to afford an absorbance value of 0.7 at 734 nm. Next, 100 μL of a sample was mixed with the working solution (3.9 mL), and the absorbance were measured at 734 nm after 20 minutes at 37 °C in the dark. Different concentrations of tolorox solution were employed as samples, with their absorbance values and concentrations being determined through their TEAC values.

### 3.7. H. Determination of Nitric Oxide (NO) Released by RAW264.7 Macrophages

RAW264.7 cells were cultured in RPMI 1640 medium containing 10% fetal bovine serum (FBS), 100 U/mL penicillin, and 100 μg/mL streptomycin at 37 °C in a 5% CO_2_ atmosphere and diluted to a final concentration of 5 × 10^5^ cells/mL. Cells (5 × 10^5^ cells/mL, 180 μL) were dispensed into 96-well plates and 20 μL test sample (IOPS dissolved in PBS diluted to final concentration of 50, 100, 200, and 500 μg/mL in the incubate mixture) were added to each well. PBS and LPS (1 μg/mL) were served as the negative and positive controls respectively. After incubation at 37 °C in a 5% CO_2_ atmosphere for 48 h, supernatants (100 μL) were collected and mixed with 50 μL Griess reagent (1% (*w*/*v*)) sulfanilamide, 0.1% (*w*/*v*) naphthylethylenediamine dihydrochloride, 2% (*v*/*v*) phosphoric acid), and incubated at room temperature for 10 min. Nitrite accumulation was used as an indicator of NO production in the medium, and nitrite production was determined by comparing the absorbance at 543 nm against a standard curve generated using NaNO_2_ [39].

### 3.8. Statistical Analysis

Results are presented as means ± standard deviation (SD). Inter group comparisons were performed by one-way analysis of variance (ANOVA) and LSD test. All of the variables were tested for normal and homogeneous variance by Levene’s test. When necessary, Tamhane’s T2 test was performed. *P* value of less than 0.05 or 0.01 is significanct and very significanct, respectively.

## 4. Conclusions

An optimum protocol ((NH_4_)_2_SO_4_, 20.16% (*w*/*v*), 9.8 mL of *t*-butanol; 30 °C; for 30 min; pH 8.0) has been identified that employs a TPP method to efficiently EY of IOPS in 2.2%; however, ammonium sulfate concentration, *t*-butanol to slurry volume ratio, and pH of slurry were found to be critical parameters for evaluation of TPP, and therefore significant to obtain maximum IOPS. Purified IOPS was shown to be a proteoglycan with a MW of 40 kDa, which was comprised of d-galactose, d-glucose, d-xylose, and d-mannose in a 2.0:3.5:1.0:1.5 molar ratios. The IOPS extract showed excellent antioxidant and immunological activities in vitro.

There is little discussion about the safety of three-phase extraction of oils, protein, and polysaccharides. Since the research is mainly for the extraction of polysaccharides from the aqueous phase, we only need focus on the ammonium sulfate mixed with water. The application of ammonium sulfate is mostly concentrated on the precipitation of proteins, which is always considered safe [40]. Studies have shown that a small amount of ammonium sulfate aqueous solution has an anti-virus effect [41], and the IOPS can undergo dialysis step to remove ammonium sulfate in this experiment. From many angles, it can be seen that the safety of extracts is beyond doubt.

The extraction and purification process of polysaccharides is a basic and popular research. Crude polysaccharide is usually extracted through classical hot water extraction and ethanol precipitation. High-purity homogeneous polysaccharide is then obtained through a series of separation and purification procedures, including deproteinization, decoloration, dialysis, and column chromatography [42]. However, these procedures are very tedious and complex, and a large amount of volatile organic solvents, such as ethanol and chloroform, are consumed. Our research is a new breakthrough in extraction of new active edible fungi polysaccharides. The TPP protocol used for the extraction of *I. obliquus* in this study could conceivably be used to extract bioactive polysaccharides from other edible and medicinal mushrooms.

## Figures and Tables

**Figure 1 molecules-24-00403-f001:**
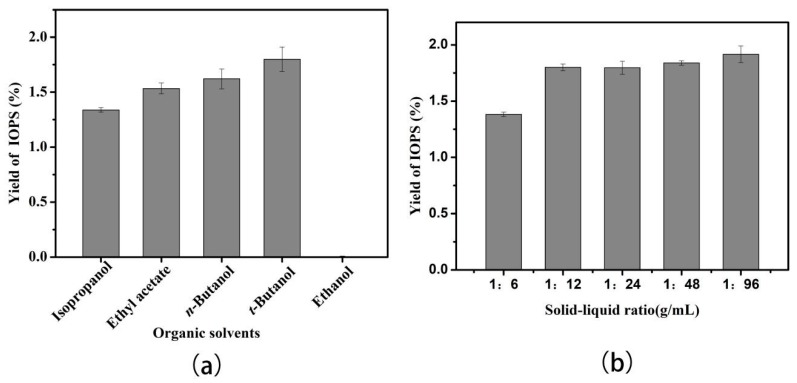
Effect of (**a**) different kinds of organic solvents on the extraction yield (EY) of IOPS using a solid-liquid ratio of 1:12, 30% mass fraction of (NH_4_)_2_SO_4_ (*w*/*v*), organic solvent (10 mL), an extraction time of 30 min, and a temperature of 30 °C at pH 8.0; (**b**) solid-liquid ratio on the EY of IOPS using a mass fraction of (NH_4_)_2_SO_4_ 30% (*w*/*v*), *t*-butanol (10 mL), an extraction time of 30 min, and a temperature of 30 °C at pH 8.0.

**Figure 2 molecules-24-00403-f002:**
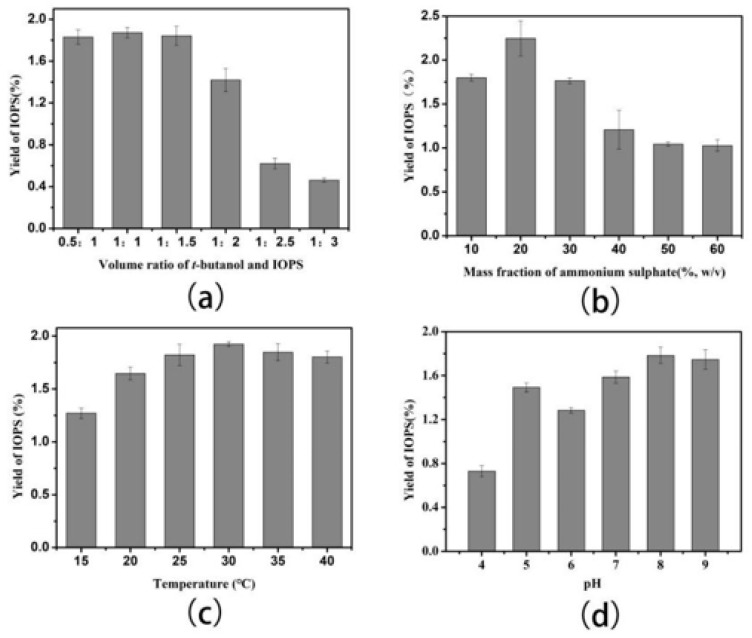
Effects of (**a**) amount of *t*-butanol (mL) on EY of IOPS using a solid-liquid ratio 1:12, a mass fraction of (NH_4_)_2_SO_4_ 30% (*w*/*v*), a time of 30 min, and a temperature of 30 °C at pH 8.0; (**b**) mass fraction of (NH_4_)_2_SO_4_ (%, *w*/*v*) on the EY of IOPS using a solid-liquid ratio of 1:12, *t*-butanol (10 mL), an extraction time of 30 min, and a temperature of 30 °C at pH 8.0; (**c**) temperature on the EY of IOPS using a solid-liquid ratio of 1:12, *t*-butanol (10 mL), a mass fraction of (NH_4_)_2_SO_4_ 30% (*w*/*v*)_,_ and an extraction time of 30 min at pH 8.0; (**d**) pH on the yield of IOPS using a solid-liquid ratio of 1:12, *t*-butanol (10 mL), a mass fraction of (NH_4_)_2_SO_4_ of 30% (*w*/*v*), an extraction time of 30 min, and a temperature of 30 °C; (**e**) extraction time on the yield of IOPS using a solid-liquid ratio of 1:12, *t*-butanol (10 mL), a mass fraction of (NH_4_)_2_SO_4_ of 30% (*w*/*v*), and a temperature of 30 °C at pH 8.0.

**Figure 3 molecules-24-00403-f003:**
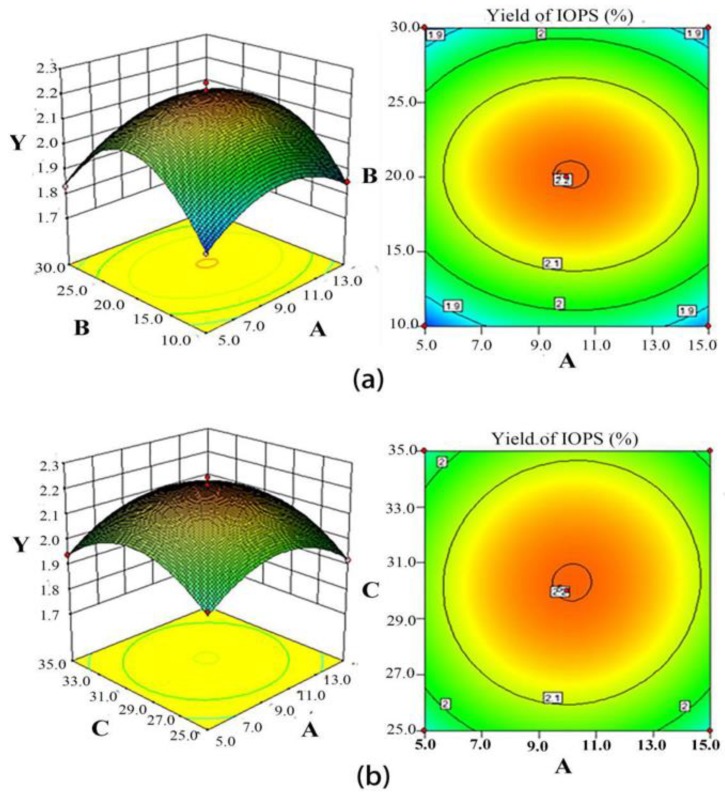
Response surface and contour plots showing (**a**) the effects of mass fraction of (NH_4_)_2_SO_4_ (A, %) and amount of *t*-butanol (B, mL) on the EY of IOPS (Y, %); (**b**) the effects of mass fraction of (NH_4_)_2_SO_4_ (A, %) and temperature (C, °C) on the EY of IOPS (Y, %); and (**c**) the effects of the amount of *t*-butanol (B, mL) and temperature (C, °C) on the EY of IOPS (Y, %).

**Figure 4 molecules-24-00403-f004:**
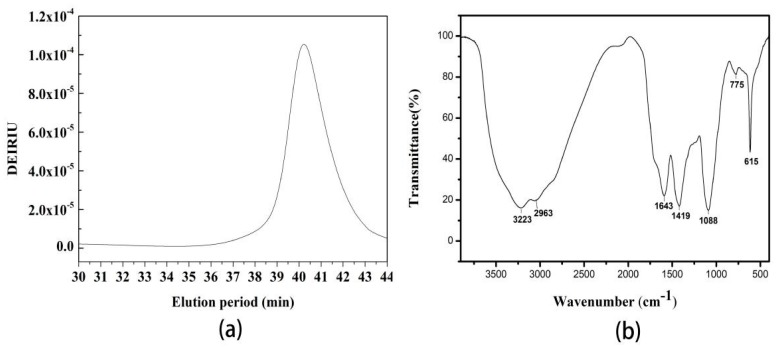
Physicochemical properties of the purified IOPS analyzed by (**a**) HPSEC-MALLS-RI chromatograms and (**b**) FT-IR spectrum.

**Figure 5 molecules-24-00403-f005:**
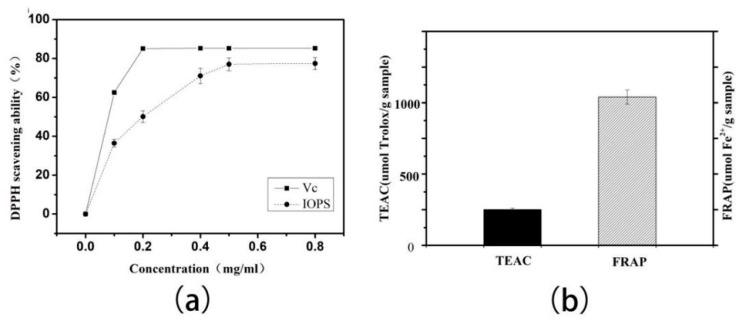
Antioxidant and immunological activities of the purified IOPS were determined by (**a**) the DPPH assay; (**b**) the TEAC and FRAP assays; and (**c**) the NO release assay, respectively.

**Table 1 molecules-24-00403-t001:** Regression coefficient estimations and significant tests used for the quadratic polynomial model used to determine the overall EY of IOPS.

Source of Variance	Sum of Squares	DF ^a^	Mean Square	*F*-Value	*p*-Value
Model	0.439876	9	0.0488751	32.91013	<0.0001
A	0.00045	1	0.00045	0.303008	0.5991
B	0.000616	1	0.0006157	0.414602	0.5402
C	0.001964	1	0.0019641	1.322553	0.2879
AB	0.000324	1	0.000324	0.218166	0.6546
AC	0.0004	1	0.0004	0.269341	0.6198
BC	0.0003	1	0.0003001	0.20209	0.6666
A^2^	0.067558	1	0.0675583	45.49052	0.0003
B^2^	0.250931	1	0.2509311	168.9649	<0.0001
C^2^	0.076877	1	0.076877	51.76525	0.0002
Residual	0.010396	7	0.0014851		
Lack of fit	0.000821	3	0.0002737	0.114326	0.9471
Pure error	0.009575	4	0.0023937		
Cor. total	0.450271	16			
			R^2^ = 0.9769	R^2^_adj_ = 0.94722	CV = 1.96

^a^ DF, degree of freedom.

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
