# Peer review of "Three-Phase Partitioning for the Extraction and Purification of Polysaccharides from the Immunomodulatory Medicinal Mushroom Inonotus obliquus"

_molecules, 2019, doi:10.3390/molecules24030403_

Round 1

Reviewer 1 Report

Another similar work has been published on the isolation of polysaccharides from Inonotus obliquus  (Xu et al. , International Journal of Medicinal Mushrooms, Volume 12, Issue 3, 2010, Pages 235-244.

1. At several points in the text, write (NH4)2SO4 instead (NH4)2SO4. Please correct them.

2.    Line 197. Please comment on the wavenumbers of FTIR spectrum (Fig. 4b).

3.  Line 235. i) How the powders were prepared? Please details; ii) 95 % v/v in water. Please correct it.

4. Line 236. Please provide the details of the ultrasound device. During the extraction the temperature was kept constant and what was the value?

5. Line 238. What are the combined extracts? Did not alcohol evaporate at the drying stage? Describe in detail the extraction with water. I think the temperature was too high and there were chemical changes. This part should be rewritten more clearly and in more detail.

6.  Line 247. Please details for “reduce pressure"

7.     Line 300. Not FeCl3·6H2O but FeCl3.6H2O

.

Author Response

Response to Reviewer 1 Comments

Another similar work has been published on the isolation of polysaccharides from Inonotus obliquus (Xu et al., International Journal of Medicinal Mushrooms, Volume 12, Issue 3, 2010, Pages 235-244.

Response: Our experiment is completely different from the work by Xu et al. Firstly; the method used to extract and purify polysaccharide from Inonotus obliquus is different. TPP is a new, efficient and safe extraction method that has never been applied to extract and purify polysaccharides from edible fungi. Although we also use response surfaces to optimize the conditions, we use hot water to extract crude polysaccharide, and use TPP to purify polysaccharide of I. obliquus. The polysaccharide content was estimated to be 57.17% after TPP (including dialysis). Xu et al. use distilled water (83oC) to extract polysaccharide of I. obliquus, and use 4 volumes of 95% ethanol to precipitate crude polysaccharides. The polysaccharide content from dry matter of culture broth (DMCB) of I. obliquus was estimated to be 41.23 mg/g by the method of Xu et al. Secondly, experimental materials are different. Xu et al. studied the dry matter of culture broth (DMCB) of I. obliquus in submerged culture, and we studied fruiting body of I. obliquus. Thirdly, we focus on the characteristics polysaccharides from I. obliquus and free-radical scavenging abilities, antioxidant activities and immunological activity in vitro, while Xu et al. concerned on antihyperglycemia, anti-lipid peroxidation and anti-oxidation effects in alloxan-induced diabetic mice.

Point 1: At several points in the text, write (NH4)2SO4 instead (NH4)2SO4. Please correct them.

Response 1: We have corrected “(NH4)2SO4” to “(NH4)2SO4”.

Point 2: Line 197. Please comment on the wavenumbers of FTIR spectrum (Fig. 4b).

Response 2: Line 197, we inserted “Fig.5b showed that purified IOPS has a typical characteristic absorption peak of polysaccharide, a strong -OH stretching vibration peak at 80575px-1, and a strong C-H stretching vibration peak of -CH3, -CH2, -CH at 2963 cm-1, a C=O asymmetric stretching vibration peak at 1643 cm-1, CH variable angle vibration peak at 1419 cm-1. Three absorption peaks at 1088, 775, 615 cm-1 indicated a pyranose characteristics of the ring.”

Point 3: Line 235. i) How the powders were prepared? Please details; ii) 95 % v/v in water. Please correct it.

Response 3: We have rewritten this section according your advice. “Fresh air-dried I. obliquus fruiting bodies were ground to powder and passed through a 10-mesh sieve. Powder of I. obliquus (50.0 g) was suspended in 500 mL of 95% ethanol in distilled water (v/v) and extracted for 1h at the frequency of 40 KHz using a SB25-12D ultrasonic generator (Ningbo Scientz Biotechnology Co., Ltd, Ningbo, China.) at room temperature to remove lipids. This operation was repeated three times. After filtration, the residue was air-dried at room temperature, suspended in 20 volumes of distilled water and extracted twice for 2 h at 100oC. The liquid extracts were combined, centrifuged (26,000 × g, 20 min, 20oC), and the supernatant was concentrated to 300 mL under vacuum that was labelled as the crude extract.”

Point 4: Line 236. Please provide the details of the ultrasound device. During the extraction the temperature was kept constant and what was the value?

Response 4: Powder of I. obliquus (50.0 g) was extracted with 500 mL of 95% ethanol for 1h at the frequency of 40 KHz using a SB25-12D ultrasonic generator (Ningbo Scientz Biotechnology Co., Ltd, Ningbo, China.) at room temperature to remove lipids.

Point 5: Line 238. What are the combined extracts? Did not alcohol evaporate at the drying stage? Describe in detail the extraction with water. I think the temperature was too high and there were chemical changes. This part should be rewritten more clearly and in more detail.

Response 5: After 95% ethanol extraction to remove lipids, the residue of I. obliquus fruiting bodies was air-dried at room temperature, suspended in 20 volumes of distilled water and extracted twice for 2h at 100oC. “Combined extracts” means that the two times liquid extracts were combined.

About “Did not alcohol evaporate at the drying stage?” there is a clerical error here. Yes, alcohol has evaporated at the drying stage; we have corrected it.

We have rewritten this section about the detail of water extraction of crude polysaccharide.

As you know, extracting polysaccharides (the cell walls of plants and fungi) using hot water 2-4 h at 100oC is very common method. These complex polysaccharides are not very digestible, so there are a few chemical changes on structure of polysaccharide 2-4 h at 100 oC. Therefore, the traditional hot water extraction method is the most commonly used method for extracting the cell walls polysaccharides of plants and fungi. ["Dietary Reference Intakes for Energy, Carbohydrate, fiber, Fat, Fatty Acids, Cholesterol, Protein, and Amino Acids (Macronutrients) (2005), Chapter 7: Dietary, Functional and Total fiber". US Department of Agriculture, National Agricultural Library and National Academy of Sciences, Institute of Medicine, Food and Nutrition Board.] Many documents used this method. For example, Xu et al. extracted the roots of P. ginseng (500 g) polysaccharide from 7.0 L distilled water at 100 °C for 4 h. [Zhang X, et al. Total fractionation and characterization of the water-soluble polysaccharides isolated from Panax ginseng C. A. Meyer. Carbohydrate Polymers, 2009, 77(3):544-552]. Fan et al. obtained crude polysaccharide from C. comatus mycelium (325.15 g) by extraction 4× with 10 vol of distilled water at 100 °C for 1 h. [Fan JM, et al. Structural elucidation of a neutral fucogalactan from the mycelium of Coprinus comatus. Carbohydrate Research, 2006, 341:1130-1134.]

Point 6: Line 247. Please details for “reduce pressure"

 Response 6: Here we changed “under reduced pressure” to “under vacuum at 45°C”.

Point 7: Line 300. Not FeCl3•6H2O but FeCl3.6H2O 

Response 7We have corrected “FeCl3•6H2O” to “FeCl3.6H2O”.

Reviewer 2 Report

The presented article deals with interesting and important issues related to the acquisition of valuable bioactive compounds from natural sources. Nevertheless, the article requires some important amendments.

The main focus is on carelessness in the editing of the text, the necessary is deletion of unnecessary spaces, Latin genre names and prefixes like t- should be written in italic and there is no lower index for numbers in some formulas of chemical compounds. Moreover, in several places (lines 52 and 354) there is no reference to the list of quoted literature. The size, description and position of all graphs should be unified. 

Additionally, several parts of the text require the Authors' comment:

1) line 65: Has the TPP method been already used for polysaccharides?

2) line 72: How can you explain the efficiency of over 100% (for tomato) for the example cited?

3) line 235: Will ethanol not also extract lipids? Would not it be better to use a different solvent, less polar at an earlier stage to remove lipids?

4) line 265: The content of the chapter indicates rather the analysis of the composition and not the properties of the polysaccharides, so the title of the chapter is inadequate to the content.

Author Response

Response to Reviewer 2 Comments

The presented article deals with interesting and important issues related to the acquisition of valuable bioactive compounds from natural sources. Nevertheless, the article requires some important amendments.

The main focus is on carelessness in the editing of the text, the necessary is deletion of unnecessary spaces, Latin genre names and prefixes like t- should be written in italic and there is no lower index for numbers in some formulas of chemical compounds. Moreover, in several places (lines 52 and 354) there is no reference to the list of quoted literature. The size, description and position of all graphs should be unified.

Additionally, several parts of the text require the Authors' comment:

Point 1: line 65: Has the TPP method been already used for polysaccharides?

Response 1: Yes. As we have discussed in conclusions section. In order to understand it cleanly, we inserted “Currently, TPP is also applied to extract and purify tapioca starch and tapioca starch derivatives [16], chitosan from shrimp shell [17], alginates from Dunaliella salina [18], levan and hydrolyzed levan from several levansucrase microorganisms, among them Zymomonas mobilis, a mobile Gram-negative bacterium [19], aloe polysaccharides [20] and Corbicula fluminea polysaccharides [21]. More recently, Wang et al. reported that TPP is utilized to separate and purify polysaccharide–protein complexes (PSP) from C. fluminea [22]. The highest extraction yield of PSP was 9.0% under the following optimal conditions: 20% (w/v) ammonium sulfate concentration, 1.5: 1.0 (v/v) t-butanol to crude extract ratio, 30 min, and 35 °C. The purified PSP also exhibited strong radical scavenging capacities and antioxidant activities in vitro. TPP have been applied in the fields of plant, animal and microorganism. However, its use for separation polysaccharides from edible and medicinal fungi has not yet been reported.” in line 65

Point 2: line 72: How can you explain the efficiency of over 100% (for tomato) for the example cited ?

Response 2: The author of the example cited explains, “The unusual enzyme recovery of 183% needs explanation. It has been often observed (Dennison & Lovrein 1997) that three-phase partitioning leads to simultaneous activation of the enzyme which (if the enzyme recovery is high) results to such an apparently observed value of 183% yield. Recently, we have found that enzyme activation frequently observed during three-phase partitioning may be the result of increased flexibility in the enzyme molecule. X-Ray diffraction studies show three-phase partitioning treated proteinase K has an unusually high B-factor (Singh et al. 2001).”

Point 3: line 235: Will ethanol not also extract lipids? Would not it be better to use a different solvent, less polar at an earlier stage to remove lipids?

Response 3: Yes, ethanol can extract and remove most of the lipids and micro molecular compounds, including colour ingredients, free amino acids, free sugars, phenols, and so on. Some organic solvents can be used to extract lipids such as petroleum ether, ethyl acetate, acetone and chloroform.  For example, Wang et al. report that C. fluminea was treated thrice with refluxing petroleum ether for 6 h each treatment to remove lipids and pigments. [WangYY, Qiu WY, Wang ZB, et al. Extraction and characterization of anti-oxidative polysaccharide–protein complexes from Corbicula fluminea through three-phase partitioning. RSC Adv., 2017, 7:11067-11075].  Shao et al. reported that 100 g of plant material from Tripterygium wilfordii was extracted with 200 mL ethyl acetate in a Soxhlet extractor for 12 h to remove lipids. [Shao D, Dunlop WD, Lui EMK , et al. Immunostimulatory and anti-inflammatory polysaccharides from Tripterygium wilfordii: comparison with organic extracts. Pharmaceutical Biology, 2008, 46(1-2):8-15]. Zhang et al. reported 100 g powder of sclerotia and 20 g powder of mycelia were defatted with diethyl ether and acetone for 4 h. [Zhang M, Zhang L, Cheung PCK, et al. Molecular weight and anti-tumor activity of the water-soluble polysaccharides isolated by hot water and ultrasonic treatment from the sclerotia and mycelia of Pleurotus tuber-regium. Carbohydrate Polymers, 2004, 56(2):123-128.].

Many documents use ethanol to remove lipids because ethanol is efficient, safe, non-toxic and cheaper. For example, Zhang et al. reported the fruiting bodies of H. erinaceus were first exhaustively extracted with ethanol under reflux for 12 h to remove lipids. [Zhang AQ, Zhang JS, Tang QJ, et al. Structural elucidation of a novel fucogalactan that contains 3-O-methyl rhamnose isolated from the fruiting bodies of the fungus, Hericium erinaceus. Carbohydrate Research, 2006, 341(5):645-649.] Fan et al. reported Dried C. comatus mycelium (325.15 g) was extracted 3× with 95% of ethanol EtOH for 1 h under reflux to remove lipid. [Fan JM , Zhang JS , Tang QJ , et al. Structural elucidation of a neutral fucogalactan from the mycelium of Coprinus comatus. Carbohydrate Research, 2006, 341(9):1130-1134.]

Point 4: line 265: The content of the chapter indicates rather the analysis of the composition and not the properties of the polysaccharides, so the title of the chapter is inadequate to the content.

Response 4: We changed the title of this chapter to “Preliminary study on the polysaccharide composition of IOPS”.

Reviewer 3 Report

The article deals with the valorization of polysaccharides from a medicinal fungus, a subject which is of present interest. Abstract, introduction, results and discussions are overall according to scientific requirements. Some issues should be however ammended:

1. lines 38-39: first sentence, referring to taxonomical information, please rephrase to: “Inonotus obliquus (Ach. ex Pers.) Pilát, also known as chaga, is a medicinal basidiomycetes fungus included in the Hymenochaetaceae family,  Hymenochaetales order, and Agaricomycetes class, respectively.”

source: Kirk P. (2013). Updated database version of Kirk PM, Cannon PF, Minter DW, Stalpers JA (2008). Dictionary of Fungi, 10th Edition, CAB International, Oxon, UK

2. line 41: lines of latitude are usually indicated by ° (degrees); please write “latitudes of 45-50° N”

3. Please write (NH4)2SO4 instead (NH4)2SO4 and correct lower case of chemical formulae throughout the text

4. line 65: Please comment on the use of the TPP method for polysaccharides.

5. line 197. Please comment on the wavenumbers of FTIR spectrum (Fig. 4b).

Author Response

Response to Reviewer 3 Comments

The article deals with the valorization of polysaccharides from a medicinal fungus, a subject which is of present interest. Abstract, introduction, results and discussions are overall according to scientific requirements. Some issues should be however ammended:

Point 1: lines 38-39: first sentence, referring to taxonomical information, please rephrase to: “Inonotus obliquus (Ach. ex Pers.) Pilát, also known as chaga, is a medicinal basidiomycetes fungus included in the Hymenochaetaceae family, Hymenochaetales order, and Agaricomycetes class, respectively.”

source: Kirk P. (2013). Updated database version of Kirk PM, Cannon PF, Minter DW, Stalpers JA (2008). Dictionary of Fungi, 10th Edition, CAB International, Oxon, UK

Response 1: We have corrected “Inonotus obliquus belongs to the eumycophyta, basidiomycotina, hymenomycetes, aphyllophorales, polyporaceeae and poriahypobrunnea petch families that are used as traditional medicinal fungus.” to “Inonotus obliquus (Ach. ex Pers.) Pilát, also known as chaga, is a medicinal basidiomycetes fungus included in the Hymenochaetaceae family, Hymenochaetales order, and Agaricomycetes class, respectively.”

Point 2: lines of latitude are usually indicated by ° (degrees); please write “latitudes of 45-50° N”

Response 2: We have corrected ° (degrees)” to “latitudes of 45-50° N”.

Point 3: Please write (NH4)2SO4 instead (NH4)2SO4 and correct lower case of chemical formulae throughout the text.

Response 3: We have corrected “(NH4)2SO4” to “(NH4)2SO4”.

Point 4: line 65: Please comment on the use of the TPP method for polysaccharides.

Response 4: We inserted Currently, TPP is also applied to extract and purify tapioca starch and tapioca starch derivatives [16], chitosan from shrimp shell [17], alginates from Dunaliella salina [18], levan and hydrolyzed levan from several levansucrase microorganisms, among them Zymomonas mobilis, a mobile Gram-negative bacterium [19], aloe polysaccharides [20] and Corbicula fluminea polysaccharides [21]. More recently, Wang et al. reported that TPP is utilized to separate and purify polysaccharide–protein complexes (PSP) from C. fluminea [22]. The highest extraction yield of PSP was 9.0% under the following optimal conditions: 20% (w/v) ammonium sulfate concentration, 1.5: 1.0 (v/v) t-butanol to crude extract ratio, 30 min, and 35 °C. The purified PSP also exhibited strong radical scavenging capacities and antioxidant activities in vitro. TPP have been applied in the fields of plant, animal and microorganism. However, its use for separation polysaccharides from edible and medicinal fungi has not yet been reported.” in line 65.

Point 5: Line 197. Please comment on the wavenumbers of FTIR spectrum (Fig. 4b).

Response 5: Line 197, we inserted “Fig.5b showed that purified IOPS has a typical characteristic absorption peak of polysaccharide, a strong -OH stretching vibration peak at 3223 cm-1, and a strong C-H stretching vibration peak of -CH3, -CH2, -CH at 2963 cm-1, a C=O asymmetric stretching vibration peak at 1643 cm-1, CH variable angle vibration peak at 1419 cm-1. Three absorption peaks at 1088, 775, 615 cm-1 indicated a pyranose characteristics of the ring.”

Round 2

Reviewer 3 Report

Amendments suggested by the reviewers were properly performed by the authors.